# An Overview of Shoreline Mapping by Using Airborne LiDAR

**Junbo Wang [1,2], Lanying Wang [2], Shufang Feng [1], Benrong Peng [1,\*], Lingfeng Huang [1], Sarah N. Fatholahi [2], Lisa Tang [2] and Jonathan Li [2]**

[1] College of the Environment & Ecology, Xiamen University, Xiamen 361102, China
[2] Department of Geography and Environmental Management, University of Waterloo, Waterloo, ON N2L 3G1, Canada
\* Correspondence: brpeng@xmu.edu.cn

**Abstract:** Since the shorelines are important geographical boundaries, monitoring shoreline change plays an important role in integrated coastal management. With the evolution of remote sensing technology, many studies have used optical images to measure and to extract shoreline. However, some factors limit the use of optical imaging on shoreline mapping. Considering that the airborne LiDAR data can provide more accurate topographical information, there are some studies that have been investigated using airborne LiDAR to map shorelines. However, a literature review that combines airborne LiDAR with shoreline measurement and extracting methods has not yet been conducted. The motivation of this paper is to present a narrative review of shoreline mapping by using airborne LiDAR, including a laser scanning system, data availability, and current extraction techniques over the past two decades. Therefore, we conducted a broad search and finally summarized more than 130 articles on airborne LiDAR technology for shoreline measurement and shoreline extraction. We find that shoreline mapping by using airborne LiDAR still meets the challenge, such as objective condition limitations, data availability limitations, and self-characteristic limitations. The current method of shoreline extraction has a great potential to be improved; particularly when combined with the emerging current state-of-the-art LiDAR point cloud processing techniques (e.g., deep-learning algorithms), they will have a brighter future. This review paper provides an overview and the current trend of shoreline mapping using airborne LiDAR, and points out the limitations, challenges, and future opportunities. Moreover, it also can serve as a starting point for novices and experts to study the shoreline mapping by using airborne LiDAR, which provides a scientific support for studying shoreline changes.

**Keywords:** airborne LiDAR; laser scanning system; shoreline mapping; shoreline extraction

## 1. Introduction

Shorelines are the physical interface that separates the land from the ocean. This interface is surrounded by a wealth of ocean resources. The desired economic development forces most coastal countries around the world to exploit these ocean resources [1]. Approximately 50% of the population around the world lives within 100 km of a shoreline, and aims to tap into these ocean resources along the shoreline for economic and social benefits [2,3]. However, natural factors and excessive human intervention have caused varying degrees of damage to coastal zones, making it somewhat vulnerable, especially the evidence that has been demonstrated in shoreline changes.

Due to the dynamic changes of shorelines depending on the temporal and spatial scale, the shoreline is not an invariable line. Thus, coastal scientists undertake studies to analyze variability and erosion-accretion trends for shorelines. Traditional measurement techniques such as ground surveys or aerial photogrammetry can be used for measuring the coastal beach erosion to track fluctuations in shoreline position [4]. However, there are some limitations of the traditional shoreline measurement, including limited spatial

resolution, large expense, time-consuming, and the requirement of a large amount of well-trained manpower [4–7]. With the progress of remote sensing technology, such as optical, microwave, and Light Detection and Ranging (LiDAR) sensors, high precision coastal maps can be obtained accurately and efficiently [4,8,9], which have been applied in practical scenarios including the generation of accurate navigation charts, the determination of marine boundaries, the monitoring of shoreline erosion and accumulation, and delineating the intertidal zones, wetlands, and other coastal ecosystems [10,11].

In terms of shoreline measurement, optical satellite remote sensing and airborne LiDAR have been widely used in modern surveying [12–19]. A comparison between optical satellite remote sensing and airborne LiDAR is shown in Table 1. Optical satellite as passive sensor technology requires daylight and optimal weather conditions for shoreline measurement. Otherwise, the collection of optical images would be affected by adverse weather conditions, such as haze, clouds, and poor light conditions [20]. Meanwhile, it may be difficult to acquire suitable images for the study area due to the revisit of time, depending on the operational cycle of the satellite [21]. In addition, most common optical images lack vertical information [22], which cannot deal with the various tide-level results. Thus, the shoreline extracted is instantaneous. When studying the trend of shoreline change, we need to correct the instantaneous shoreline using the tidal level data. In addition, different coastal types also can affect the accuracy of optical satellite imagery-derived shorelines [23].

**Table 1.** Comparison of airborne LiDAR and optical satellite remote sensing.

| Characteristic | | Airborne LiDAR | Optical Satellite |
|---|---|---|---|
| System | Sensor technology | Active | Passive |
| | Spatial Resolution | Relatively High | Relatively Low |
| Performance | Dataset scale | National and regional | Global |
| | Vertical accuracy | High | Low |
| | Temporal resolution | Low | High |
| | Lighting conditions | Day and Night | Daytime |
| Operation Restriction | Cloud condition | No impact | Impact |
| | Terrain Condition | No impact | Impact |
| Data | Type | Point cloud | Raster Imagery |
| Mapping Quality | Spatial information | Three-dimensional | Two-dimensional |
| | Shoreline extraction | Directly extract from point cloud or Image-processing techniques | Visual interpretation or image-processing techniques |

In contrast, airborne LiDAR, as an active remote sensing technology, has the advantage of high speed and a high precision of measurement, and it is not affected by lighting conditions [24]. More importantly, the real-time three-dimensional spatial information can be highly automated and captured to support rapid deployment in the coastal zone, due to the ability of LiDAR allowing for the observation of targets without terrain limitation [12,25]. For example, the U.S. Army Corps of Engineers has developed the SHOALS project to map coastal areas in a large-scale region [26], and has developed the Coastal Zone Mapping and Imaging LiDAR (CZMIL) system to improve survey efficiency and system reliability in coastal areas [27]. Canada implemented a national LiDAR project called FU-DOTERAM to assess topographic and bathymetric elevation characteristics in 2006 [28]. The latest news we collocted is that National Oceanic and Atmospheric Administration (NOAA) has provided over 800 new coastal LiDAR datasets covering 1.4 million $km^2$ in 2022 (https://www.opentopography.org/news/noaa-coastal-lidar-data-now-available-academic-users-through-opentopography (accessed on December 8, 2022)). Moreover, LiDAR technology has another advantage that can be used as a standalone measurement device or that can combine with with other remotely sensed devices. However, it is a pity that the temporal resolution of airborne LiDAR is closely related to the project requirements,

where the operation time depends on the project period (Table 1). It is lower than that of optical satellite remote sensing, which operates all day long and visits regularly, such as SPOT [14].

The investigation of shoreline changes is one of the main applications using remote sensed devices in coastal mapping. However, shorelines changes over time with the tidal and complex coastal environment increase the difficulty of mapping in coastal areas. The shoreline extracted from the airborne LiDAR data provides an efficient, highly accurate, useful, and fast solution [29,30]. This is because it provides not only more accurate geometrics information for the topography, but it also provides a richer radiometric information such as the intensities from different channels, which help to better distinguish between the water and land [24,31]. Although airborne LiDAR demonstrates the good performance of shoreline measuring according to some studies [32–34], its main drawback is that it cannot directly obtain the positions of shorelines due to it storing and organizing the data as unordered discrete points which are also called point clouds [12]. Thus, how to appropriately adopt and accurately use the airborne LiDAR data for shoreline extraction is an important topic for monitoring shoreline changes, which is also the motivation of writing this review article.

Indeed, this topic has been researched since the late 1990s [35], while unfortunately, an overview of shoreline extraction from airborne LiDAR point cloud data has not been seen. Meanwhile, there are also some questions that have not been answered, such as what sensor could be used, which dataset can be assessed, what technologies have been available, and what opportunities and challenges will be faced in the future for shoreline extraction. Therefore, the objectives of this review paper are to:

1.  Provide an overview and the general trend of airborne LiDAR systems used in shoreline mapping.
2.  Review in detail the current approaches for mapping the shoreline from airborne point clouds.
3.  Identify the limitations and challenges for shoreline mapping using airborne LiDAR, and provide future potential directions for this topic.

## 2. Review Approach

This paper collects the publications of shoreline extraction performed on airborne LiDAR data, including both the instantaneous shoreline directly retrieved from point clouds, and the "true" shoreline position calculated based on the LiDAR-derived DEM and shoreline indicators (more details will be provided in the Section 4) according to the time span between 2000 and 2022. We not only focused on the peer-reviewed research articles, but we also reviewed the technical reports released by the organizations from the government.

Our search was based on three groups of keywords linked with the operator AND, and within the group, we used the keywords with the operator OR: (a) Keywords of the shoreline included shoreline, coastline, shore area, and coastal area; (b) Keywords of applied tasks included mapping, extracting, extraction, measure, monitoring, and management; and (c) Keywords of LiDAR systems included airborne LiDAR, airborne laser scanning, airborne laser topographic, and bathymetric scanning.

There are a total of 1436 publications that are related to shoreline and airborne LiDAR that were collected by searching the database of Google Scholar and Web of Science. Firstly, we removed 632 duplicated articles. After that, the we selected the rest of the literature, carefully based on the relevance of the title and abstract. Sometimes, we accessed the full paper to determine the selection. Then, we actively considered the studies regarding sensors from different airborne LiDAR, and used different methods to extract the shoreline from airborne LiDAR data during the examination progress. Meanwhile, the literature was excluded that: (a) focused only on shoreline or LiDAR, (b) shoreline extraction only from remote sensing image without fusion with airborne LiDAR data, and (c) the studies not involving the extraction of shorelines from airborne LiDAR data.

After irrelevant literature were excluded, a total of 134 studies were selected according to a high degree of relevence that linked with shoreline extraction derived from airborne LiDAR. To help the readers to more easily capture the whole image of shoreline mapping via the airborne LiDAR system, we organized the review article into the following sections. In particular, the airborne LiDAR systems development, the dataset availability, shoreline mapping methods, and current limitations and challenges of airborne LiDAR for shoreline mapping are reviewed and discussed. We also provide some state-of-the-art methods of LiDAR point cloud processing which have the potential to improve the shoreline mapping domain. Finally, several promising directions for future shoreline mapping by using airborne LiDAR are highlighted.

## 3. Airborne LiDAR Systems Development and Datasets Availability for Shoreline Mapping

Airborne LiDAR consists of a series of components, of which there are three core pieces of equipment, namely, laser scanning system, differential GPS/GNSS, and Inertial Measurement Unit (IMU). This system allows for the instant collection of a three-dimensionality (3D) point cloud by capturing the reflectance energy emitted by the sensor [36–38].

As the most important equipment of airborne LiDAR, the laser ranging system makes it easy to be operated both day and night, which undertakes the role of sending and receiving laser signals. It can detect and record laser energy in different ways, including discrete-return, full-waveform, and photon-counting. Although the theory of laser was first introduced by Townes and Schawlow in 1958 [39]; it was not widely used until the 1990s. This development allows airborne LiDAR as an ideal choice for measuring shoreline environmental parameters, especially for the topography, ground objects, and vegetation classification in both surface and submerged areas [40]. More stable data of shoreline measurement (e.g., datum-derived contours) from airborne LiDAR are obtained than in High Water Line (HWL) measurements from aerial photographs, which benefit by being not subject to the effects of short-term fluctuations in wave energy and water level [41].

### 3.1. Airborne Laser Topographic and Bathymetric Scanning System for Shoreline Measurement

Currently, there are two major types of airborne LiDAR systems that are commonly used in shoreline area surveys, including the airborne laser topographic scanning system (ALT) and the airborne laser bathymetric scanning system (ALB). Tables 2 and 3 summarized the development of airborne laser scanning systems, including ALT and ALB, which were used in shoreline mapping and monitoring applications.

With the development of the ALT, the accuracy of the data has been improved (Table 2). Many studies have confirmed that shoreline extracted from the ALT can actually deliver a satisfied and accurate shoreline position, and also be suitable for larger scale shoreline extraction projects [42–45]. Recently, the Optech Pegasus has updated a sensor with horizontal and vertical accuracy, respectively, at 16 cm and 5–20 cm, at 1200 m Above Ground Level (AGL), providing better detection results between the ground and the water surface [46,47].

Compared to traditional measurement techniques, shoreline positions generated using ALT are sufficiently accurate and support further analysis [48–52]. They have the ability to collect the cross-environmental profiles of coastal topography [12,39,53,54]. White et al. [51] introduced that there are less requirements for the tide window to conduct shoreline surveys using the ALT. Moreover, practices have proven that the ALT can easily approach areas that are difficult to access [55]. This capacity has made it possible to conduct annual high-resolution shoreline surveys [56]. However, there are some cases that show that ALT may not be able to make some detections in specific areas, such as immersed lands, jetties, and very shallow water [50].

The ALT provides very detailed terrain information, but its lasers cannot penetrate the water surface itself. Considering the features of coastal areas, and with the development of

airborne LiDAR technology, it was able to integrate more advanced sensors for collecting more accurate and informative point clouds from different wavelengths, such as Optech CZMIL and Riegl VQ-880G for measuring shoreline characteristics [24,57,58]. The ALB commonly employs two laser rangefinders with different wavelengths, near-infrared (NIR) and green [59], as shown in Table 3. The measuring principle of the ALB is the NIR beam, which can measure the land and water surface [59], and the green beam can penetrate the water [60] and is reflected back from the seabed or lakebed, as shown in Figure 1. The measuring depth can vary over a range of 25–70 m, depending on the different systems [57]. ALB is becoming a fundamental tool for coastal scientists within coastal studies, due to its bathymetric ability to distinguish topographic (high density) and bathymetric (low density) LiDAR points, providing the elevation data that are critical to producing datum-based shoreline, and that are suitable for measuring the features in coastal areas [13,25].

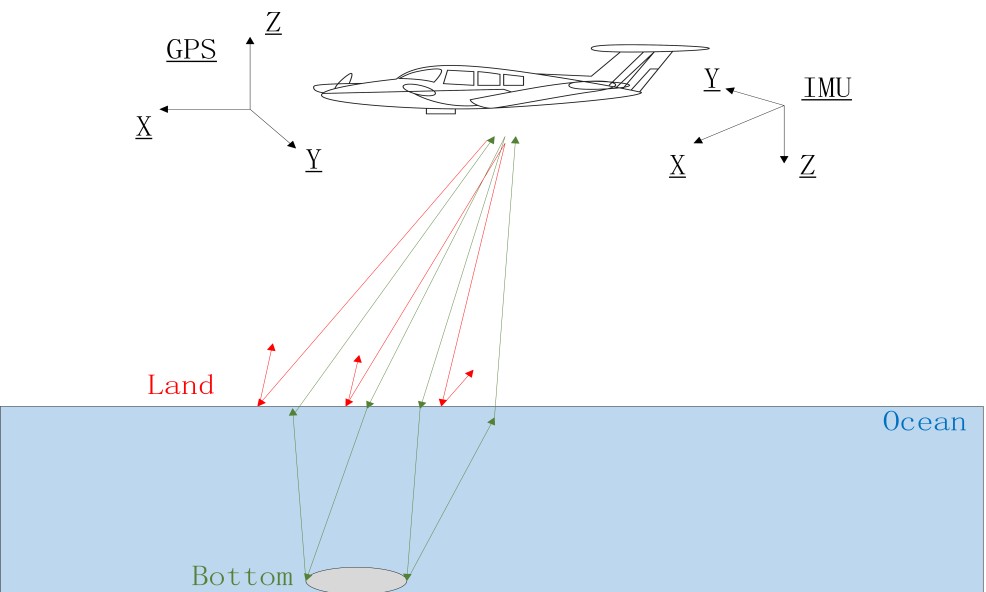

**Figure 1.** Diagram of waveform reflected from ALB.

With the requirements of both the academic and industrial fields, ALB is being constantly updated and upgraded. The upgraded ALB makes a great contribution to improving the accuracy of shoreline measurement. Since 2000, many researchers have proposed various methods and sensors to perform shoreline measurements by using ALB (Table 3). For instance, NASA has built a new ALB called "EAARL" in 2001, which uses a 532 nm green laser with full waveform and an across-track scan pattern, and an RGB digital camera and a color infrared multispectral camera installed [61]. Some researchers have used EAARL-generated datasets to extract a shoreline without tide coordination [62–64], and the result was demonstrated to have better performances than ATM [62].

Furthermore, commercial companies (e.g., Optech, Riegl, and Leica) keep designing and upgrading their products (Table 3). Experts have successfully conducted shoreline extraction and monitoring studies using datasets collected by CZMIL [15,27,65]. More advanced ALB have been engineered by more companies, such as Leica Chiroptera II (Leica Geosystems AG, Switzerland) and Riegl VQ-880G (RIEGL Laser Measurement Systems GmbH, Austria). Webster [58] indicated that the Leica Chiroptera II has the ability to monitor subtle changes in coastal areas. Madore et al. [66] have tested that the Reigl VQ-880G can successfully aid with large-scale coastal surveys, providing more seamless datasets for transitional areas between topography and bathymetry.

*3.2. Datasets Availability in Coastal Areas*

Under the environment of more and more governments, organizations, companies willing to share, and open data, more and more countries provide the airborne LiDAR data online. After broadly searching, we summarized some available datasets in Table 4. The main resources of this table are from OpenTopography (https://www.opentopography.org/ (accessed on December 8, 2022)), and the government websites. These datasets are usually initiated by national or regional projects, and are conducted by the lower-level municipal governments or organizations such as provinces or states. For example, the province New Brunswick is in the eastern coastal area of Canada, and Airborne Lidar data were collected from 2015 to 2018 within the province, which fully covered the coastal area of the whole province and is open access. (https://geonb.snb.ca/li/ (accessed on December 8, 2022)).

**Table 2.** A summary of the airborne laser topographic scanning system (ALT) used in shoreline-related studies.

| Year | Sensor | Laser Range | Pulse Repetition Frequency | Vertical Accuracy | Horizontal Accuracy | Operation Altitude | Related Studies |
|------|--------|-------------|----------------------------|-------------------|---------------------|--------------------|-----------------|
| 1996 | ATM | 1064 nm | 10 kHz | 0.15 m | 0.8 m | Typically 400–800 m | Coastal mapping and monitoring [13,42,45], shoreline extraction [43,44] |
| 1998 | Optech ALTM 1210 | 1100 nm | Max 10 kHz | 0.15 m | 0.8 m | Up to 1.2 km | Shoreline mapping [41] |
| 1999 | Optech ALTM 1225 | 1024 nm | Max 25 kHz | 0.15 m | | Up to 2 km | Coastal application [67] and shoreline extraction [56,68] |
| 2000 | Optech ALTM 1233 | 1100 nm | Max 33 kHz | | | | Coastal application [67], Shoreline changes and features extraction [32], Beach segmentation [69], Inland water boundary extraction [21] |
| 2002 | Optech ALTM 2050 | 1064 nm | Max 50 kHz | 0.15 m (1200 m AGL) | | Up to 2 km | Shoreline mapping [62,70] |
| 2003 | Optech ALTM 30/70 | 1064 nm | Max 70 kHz | 0.15 m (1200 m AGL) | $1/2000 \times$ altitude ($1\sigma$) | Up to 3 km | Shoreline mapping [55], coastal erosion and accretion [71] |
| 2004 | Optech ALTM 3100 | 1064 nm | Max 100 kHz | 0.15 m (1200 m AGL) | $1/5500 \times$ altitude ($1\sigma$) | Up to 3.5 km | Coastal mapping [50] and shoreline extraction [49,51] |
| 2008 | RIEGL Q680i-D | 1550 nm | Max 400 kHz | 0.02 m ($1\sigma$) (250 m AGL) | | Up to 1.6 km | Shoreline extraction [72,73] and volumetric changes of soft cliff coast [74] |
| 2012 | Optech Pegasus HA500 | 1064 nm | Max 500 kHz | 0.05–0.2 m ($1\sigma$) | $1/7500 \times$ altitude ($1\sigma$) | Up to 5 km | Shoreline extraction [46,47] |

Some parameters of ALT are referenced from Shan and Toth [57], Bakuła [75], García-Quijano et al. [76], Pfennigbauer and Ullrich [77].

**Table 3.** A summary of airborne laser bathymetric scanning system (ALB) used in shoreline related studies.

| Year | Sensor | Laser Range | Pulse Repetition Frequency | Depth Accuracy | Vertical Accuracy | Horizontal Accuracy | Operation Altitude | Related Studies |
|------|--------|-------------|---------------------------|----------------|-------------------|---------------------|--------------------|-----------------|
| 2001 | EAARL | 532 nm | 3–10 kHz | | 5–10 cm | <1 m | Nominal 300 m | Shoreline mapping [62,63], coastal monitoring [64] |
| 2003 | Optech SHOALS 1000T | 532 nm + 1064 nm | Max 10 kHz | $\sqrt{0.5^2 + (0.013 \times depth)^2}$ m | | 2.5 m (1$\sigma$) | 200–400 m | Seafloor mapping [78], shoreline mapping [62] |
| 2006 | Optech SHOALS 3000T-H | 532 nm + 1064 nm | 20 kHz | 0.25 m (1$\sigma$) | 0.25 m (1$\sigma$) | 2 m (1$\sigma$) 1/500 × altitude (1$\sigma$) | 300–400 m | Coastal mapping [79] and shoreline extraction [80] |
| 2010 | Optech CZMIL | 532 nm + 1604 nm | 10 kHz (green), 70 kHz (infrared) | $\sqrt{0.3^2 + (0.013 \times depth)^2}$ m, 2$\sigma$, 0–30 m | 0.15 m (2$\sigma$) | 1 m (2$\sigma$) | Nominal 400 m, up to 1 km | Coastal mapping and monitoring [15,27,65] |
| 2015 | Leica Chiroptera II | 515 nm + 1064 nm | 35 kHz (green), 500 kHz (infrared) | 0.15 m | 2 cm (1$\sigma$) | 0.20 m (1$\sigma$) (400 m AGL) | 400–600 m, up to 1.6 km | Coastal mapping [58] and shoreline monitoring [81]) |
| 2018 | Riegl VQ-880G | 532 nm + 1064 nm | Max 550 kHz | $\sqrt{0.3^2 + (0.013 \times depth)^2}$ m | 10 cm | | Max 800 m | Coastal mapping [66] |

**Table 4.** A summary of available airborne LiDAR datasets in coastal area.

| Country | Data Format | Spatial Resolution | Surveyed Year | Coverage | Additional Note | Reference |
|---|---|---|---|---|---|---|
| Australia | Airborne LiDAR-derived DTM | 5 m | 2001–2015 | 45,000 km$^2$ | Cover Australia's populated coastal zone; floodplain surveys within the Murray Darling Basin, and individual surveys of major and minor population centers. | https://www.ga.gov.au/ |
| Canada | Airborne LiDAR point clouds | 1–2 m | 2013–present | Partially covered eastern coastal area and Great Lakes area | Provincial-based nationwide project covering most major cities. | https://open.canada.ca/ |
| Japan | Airborne LiDAR point clouds | | Just launched | 35,000 km of coastline | Map of the Sea Project launched in 2022 | https://www.jha.or.jp/en/jha/ (accessed on December 8, 2022) |
| Scotland | Airborne LiDAR point clouds | 4 points/m$^2$ | 2011–2021 | More than 45,078 km$^2$ in total area | 5 phases, covering the partial coastal area | https://remotesensingdata.gov.scot (accessed on December 8, 2022) |
| USA | Airborne LiDAR point clouds | 0.15–3 m | 1999–present | Fully covered inland USA coastal area and Great Lakes area, partially covered Alaska | Surveyed by the U.S. Army Corps of Engineers, NOAA, and U.S. Geological Survey | https://coast.noaa.gov/digitalcoast/data/jalbtcx.html (accessed on December 8, 2022) |
| New Zealand | Airborne LiDAR point clouds | 1 m | 2010–present | More than half coastal line | Still ongoing to collect the data | https://www.linz.govt.nz/products-services/data/types-linz-data/elevation-data/provincial-growth-fund-LiDAR-data-collection-now-progress (accessed on December 8, 2022) |

Another example is the Scotland LiDAR project, which was conducted for five phases (still ongoing). Phase I was initialed by the Scottish Government, Scottish Environmental Protection Agency (SEPA), and Scottish Water, collaboratively. Phase 3 was initially captured by Fugro for the Scottish Power Energy Network (www.spatialdata.gov.scot (accessed on December 8, 2022)). The collaboration between the cross-functional organizations to collect the dataset may occur because the cost of collecting airborne LiDAR data is relatively high. Similarly, we have also noticed that well-developed countries such as the USA, where LiDAR data have fully covered the coastal area nationwide, have more open airborne LiDAR-based data of the coastal area. Therefore, even though the airborne LiDAR has demonstrated the ability and advantage in shoreline mapping tasks, data availability is still a challenge.

## 4. Comparing Shoreline Extraction Methods from Airborne LiDAR Data

### 4.1. Shoreline Indicators

As a key step in shoreline monitoring, extracting the shoreline position plays an important role for understanding shoreline changes. Since the coastal zone environment is complex, shorelines to be extracted automatically, efficiently, and accurately have been investigated by many studies. However, there is a question that needs to be defined before extracting a shoreline, what is the shoreline, before extracting the shorelines.

Theoretically, the shoreline has been defined as the continuous boundary between the water body and the land. However, this definition in practice is a challenge to apply. This is because this boundary can vary over a wide diversity of indicators and changes over time, such as tidal level and geomorphologic aspect [2,4]. In practice, scientists often use shoreline indicators as a proxy feature to represent the "true" position for the shoreline [26], because shorelines are only meaningful when they are studied from the spatiotemporal sense [2], otherwise it is just a line. Boak and Turner [4] list 45 shoreline indicators to promote the shoreline to be extracted, and categorizes them in three features: (a) features that can be manually interpreted from the remotely sensed data (e.g., dry/wet line, HWL), (b) features based on tidal datum indicators (e.g., Mean High Water (MHW) or Mean Lower Low Water (MLLW)), and (c) features extraction from remotely sensed data using algorithms such as classification and segmentation techniques. These features can be selected according to the research purpose, available data sources, and other factors related to the target. Toure et al. [2] summarized seven types of indicators to use in coastal monitoring and survey, that can reflect the environment modification in coastal areas based on the study results of Boak and Turner [4], and argues that it is not necessary to be too sensitive to local conditions for fluctuations [2].

Based on the literature of what we collected, there are two kinds of shoreline indicators that can be extracted from the airborne LiDAR data. One is the boundary between the water and land at the captured time such as the instantaneous water line [30,48]. Another is the line as a proxy shoreline feature (e.g., MHW) derived from the Digital Elevation Model (DEM) with a high resolution and high quality [41,43,44]. The following section focuses on the specific shoreline extraction methods.

### 4.2. Shoreline Extraction Methods

As one of the most efficient and reliable technologies for collecting terrain data, airborne LiDAR has become a fundamental application for mapping the shoreline [43]. However, airborne LiDAR can be used as a data source for shoreline extraction, but the "true" position of shoreline features (e.g., MHW, MLLW) cannot be visually interpreted from LiDAR data using the human eye [26].

Therefore, according to Section 4.1, it is presented that the shoreline feature can be extracted using different processing technologies; we summarized different shoreline extraction methods by mainly categorizing them into two groups (Table 5), which respectively involve the shoreline indicators that we have distinguished. One cost-effective way to capture the changing morphology of the shoreline is to obtain a proxy shore-

line feature (e.g., MHW) via DEM derived from laser scanning provided by airborne LiDAR [9,82,83]. Another approach is to extract the boundary (water or land) directly from that generated from the point clouds data at the captured time using classification techniques [30,48]. Figure 2 directly presents the process of shoreline mapping by using airborne LiDAR. We note that the shoreline can be extracted based on the generation of DEM or not, no matter what shoreline indicator was selected. In the early stage of studying shoreline extraction, shoreline extraction methods based on DEM generation were the main methods from 2002 to 2010. Subsequently, a trend of diversification have been demonstrated for the shoreline extraction methods. Table 5 also provides a comparison of these methods. However, these methods have both advantages and disadvantages, and they need to be chosen according to the specific object of the study. In addition, using the pre-processing method (e.g., noise reduction and correction) to improve the quality of shoreline extraction is necessary, but it is not the priority in this paper, and related literature can be referred to for details [46,84,85].

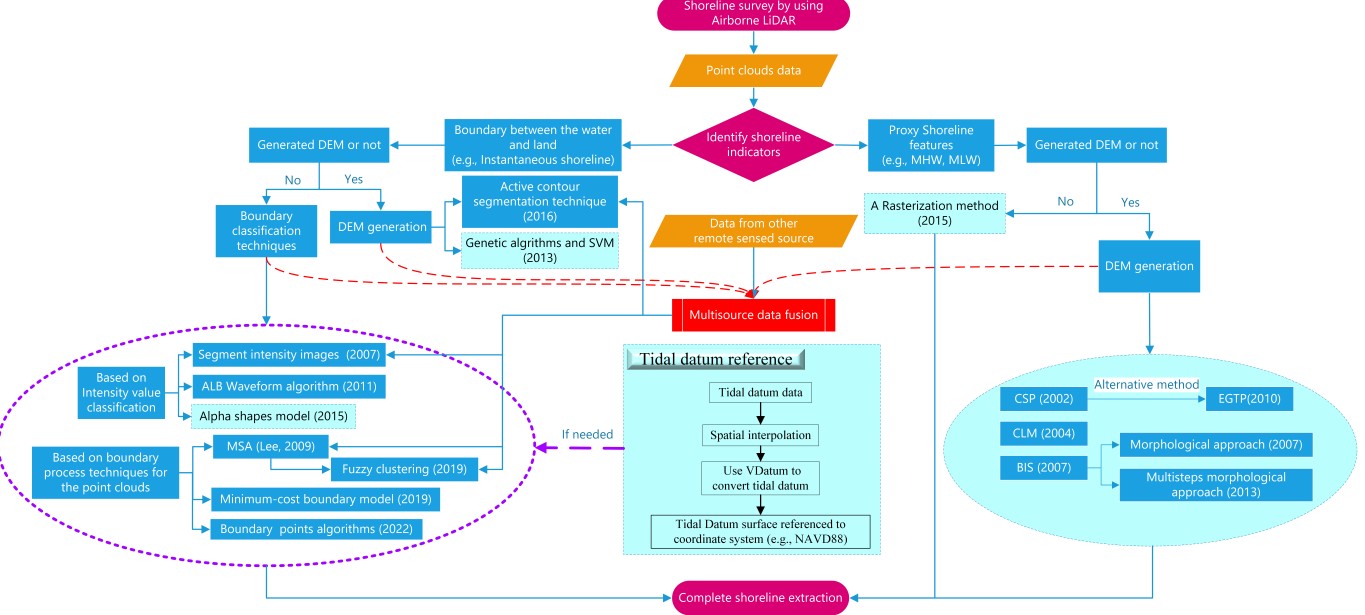

Notes: The dark cyan box indicates that it is necessary to refer to tidal data during the process of shoreline extraction.

**Figure 2.** Flow diagram of the mainstream shoreline extraction method.

**Table 5.** Compared the representative method of shoreline extraction.

| | Methods | Source Data | Pro | Cons | Accuracy | Horizon Error | Vertical Error | Shoreline Indicators/Types/Features |
|---|---|---|---|---|---|---|---|---|
| | CSP [44] | DEM generated from LiDAR Point cloud | Rapid estimation of objective, Large-scale coastal change extracts any elevation datum or elevation-based definition of shoreline after profiles have been created | Tedious and time-consuming to analyze individual profiles, closed profiles spacing, high tides, large waves, storm surge, and run-up may obscure the location of the vertical datum | 1.5 m (horizon) | 0.42 m | 0.15 m | Shoreline at low tide/Sandy shorelines/MHW |
| | EGTP [86,87] | | More robust, more independent, lower percentage of transects lost, less sensitive to noise and outliers, more continuous way and efficient to represent shorelines and complex shapes, less labor than CSP | | | | | Mean sea level, curved and closed coastal features |
| | CLM [41] | | Easy, high accuracy, no considered transect, profiles | Numerous manual editing, low efficiency, curve fragmentation | | | 0.2 m | HWL, MHW, MHHW |
| | Original method [43] | | Avoiding curve fragmentation, reduce manual editing, high accuracy | Complicated DEM construction, time-consuming, inherited errors from DEM construction, only extracted a certain tidal datum | 4.5 m (horizon) | 0.8 m | 0.15 m | MHHL |
| Binary image segmentation | Upgraded method [52] | Image (land and sea) generated from DEM derived from LiDAR point cloud | More robust, efficient and universal, less labor | | | 0.9 m | 0.18 m | High/Low tide lines |
| | Morphological operators [73] | | Easy to operate, significantly decreased influence of artificial structures on the shoreline extraction | | | | | MHW |

**Table 5.** *Cont.*

| | Methods | Source Data | Pro | Cons | Accuracy | Horizon Error | Vertical Error | Shoreline Indicators/ Types/Features |
|---|---|---|---|---|---|---|---|---|
| Multisource approach fusion | Several image-processing algorithms [42] | Hyperspectral images and LiDAR point cloud | Solve the coastal mapping complexity | | 0.1 m (vertical) | | | Instantaneous shoreline |
| | MSA [48] | | High accuracy for man-made structures | Only extracts to instantaneous shorelines | 2 m (horizon); 0.3 m (vertical) | | | |
| | Based on tidal estimation [88] | Aerial images and LiDAR point cloud | Obvious effect to extract sandy and rock shoreline | Poorly effective at extracting argillaceous shoreline | | | | HWL/sandy/rock/ muddy shorelines |
| | SVMs [72] | | Effective, higher accuracy, better performance | | 1 m (horizon); 0.15 m (vertical) | | | MHW/MLW/MLLW |
| Intensity-based method [32] | | Intensity value | Robust system, can exploit natural geometry of shorelines | | | | | Beach |
| ASM [89] | | | Fast, stabilized, adaptive, irregular polygons, not need build triangulation or shorten | Manual small target for inerratic shoreline, easy to misjudge (such as ships point clouds) | | | | |
| Fuzzy clustering [47] | | SAR image and LiDAR intensity values | Parameters without defined by users | Data quality would be decreased | 0.7 m (Vertical) | | | |
| A rasterization method [90] | | | Decline error, convenience, highly efficient, extracts other supplementary shorelines without DEM constructed | | | | | Mean High Water Springs |
| Minimum-cost boundary model [30] | | LiDAR Point cloud | Without any imagery information and human interaction, minimum-cost model | | | | | |
| Boundary points algorithm [91] | | | Suitable for shoal and muddy shorelines, robust | Non-automatic extraction | | | | Shoal and muddy shorelines |

### 4.2.1. Based on Proxy Shoreline Features

Proxy shoreline features refer to shorelines extracted from the LiDAR-derived high-accuracy and high-density DEM. DEM usually can be represented through numerous models, such as the regular grid (usually square grid), the triangular irregular network (TIN), and the contour line model [92]. These models can be generated from different interpolation methods, including Inverse Distance Weighted (IDW) or Kriging.

The study of coastal areas using DEM generated from LiDAR data can be traced back to the late 1990s. Woolard and Meredith have investigated the topography changes in North Carolina, which are based on DEM generated from LiDAR data, including changes of sand dunes, and shoreline changes caused by hurricanes [93,94]. However, their results cannot fully represent and understand the terrain characteristics of beaches and sand dunes, and the morphological changes of shorelines, due to their DEM resolution being too coarse [83]. However, these studies provide a good idea for extracting shoreline from DEM. Subsequently, these methods of shoreline extraction from DEM generation have attracted the interest of many experts to explore [41–43,73,95,96]. The representative methods are including the cross-shore profiles (CSP), contouring line method (CLM) and the binary image segmentation (BIS).

Utilizing these methods to obtain shoreline position is inferred from elevation values provided by the LiDAR-derived DEM, the values of which need to be referred to the tidal datum surface [72,84,97,98]. This can overcome biases or horizontal shifts for the positions of shorelines caused by different tidal levels [99], which are associated with using a wet/dry line (visual feature) on aerial photographs as a proxy shoreline feature [26]. However, due to the existence of different reference systems in practice, LiDAR data and tidal datum should be converted to a common coordinate system (e.g., NAVD 88) by using VDatum. The newest version is VDatum 4.5.1. Details can be found in NOAA (https://coast.noaa.gov/digitalcoast/tools/vdatum.html (accessed on December 15, 2022)). As a vertical datum transformation tool developed by NOAA, VDatum allows users to vertically transform geospatial data among a variety of ellipsoidal, orthometric, and tidal datums [100]. VDatum provides a bridge to link DEM and tidal datum, which becomes the cornerstone of new ways to obtain shoreline information and supports efficient coastal management [98].

On the other hand, using these methods of extraction that are based on DEM may have issues such as the filtering of LiDAR data, interpolation methods, the resolution of DEM, and others, which still need to be paid more attention to during DEM generation [85]. For example, the accuracy of DEM decreases when the LiDAR point clouds are interpolated into a regular grid using the DEM segmentation method [101]. It is worth to note that the DEM precision is also affected by the spatial resolution and the terrain complexity. Especially in coastal zones, it can easily cause geometric distortion and large shoreline extraction errors, due to the abrupt topography changes. Dong et al. [101] argues that the conventional interpolation algorithms for DEM generated cannot retain topographic features. They proposed a method to extract shorelines with topographic constraints from LiDAR point clouds data [101]. The method reduces the errors when interpolating regular grids by constructing constrained Delaunay triangles. It also accelerates the shoreline smoothing problem associated with the elevation correction of the airborne LiDAR point clouds [101].

**The Cross-Shore Profiles**

At the beginning stage of studying on shoreline extraction from airborne LiDAR, an objective technique called the Cross-Shore Profile (CSP) was first developed to extract shoreline position based on the profiles of DEM derived from airborne LiDAR using Stockdon [44]. The main step is the use of a linear regression to fit the elevation information of a point cloud on each foreshore profile. The position of the shoreline is identified by connecting the shoreline points on the individual profiles, which is the intersection of the water levels and regression lines. However, some errors should still be considered and minimized while using the CSP method. For example, due to the sea wave run-up, it may contaminate the airborne LiDAR data in the foreshore area [44].

At present, the CSP method has been the basis of many studies conducted on shoreline extraction [45,49,102–108]. However, a disadvantage of the CSP method is its tediousness and how it is time-consuming during the analysis process, especially for analyzing the closer profile spaces [43,109]. Therefore, the CSP method has been modified and upgraded by some researchers to adapt to the different requirements. For example, Ruggiero and List [110] provide a CSP-based method to evaluate proxy datum shorelines bias on a regional scale for shorelines change research, which was first time for evaluating the uncertainty of the HWL caused by water level fluctuations. This has obvious implications for the study of shoreline change at long time and spatial scales, due to the quantification of the uncertainty of the shoreline position [110].

Compared to the CSP, Aguilar et al. [86] and Luque et al. [87] strongly recommend the use of the Elevation Gradient Trend Propagation method (EGTP) using an iterative grid-based data technique, as an alternative to the CSP method for mapping shorelines. EGTP shows a lower percentage of transects lost, and it is more robust. Moreover, it is less sensitive to noise and outliers, and performs well when dealing with very curved and even closed coastal features. Its implementation steps are firstly to extrapolate grid points of unknown heights using the elevation gradient trend of every grid point computed. Until the new grid points just exceed or are below the selected tidal datum, the process would be stopped and repeated. Next, the shoreline is plotted by connecting the boundaries that separate the grid points, until the complete shoreline is mapped.

**The Contouring Line Method**

Robertson et al. [41] developed a simple method called the contouring line method (CLM). It was a similar approach to the CSP, but the difference is that the CLM is a method based on local tide levels to track specific elevation to extract the shoreline. Shorelines can be contoured from a zero value derived from the water line, which are obtained from the results of subtracting the tidal datum from DEM. This method is easily implemented by maintaining a high extraction accuracy and allowing for an analysis of the entire shoreline position without the transect profiles [41,52]. Moreover, its extracted results are more rigorous than the wet/dry line of shoreline indicators from aerial photography, due to the elimination of the ambiguity present in the boundary [68,96,111,112]. This method produces excessively noisy or broken shorelines during the contouring process. The processing results should be corrected because of the vertical error that is derived from coastal terrain (slopes beach) measurement [41]. Therefore, manual editing and re-digitizing are required to dispose of erroneous shoreline in order to keep and to refine the shorelines [41,96,97].

The CLM was used by a number of experts because it was widely employed in GIS and mapping software packages [51,55,56,95]. However, the method still has the potential to be optimized during the process of shoreline extraction [43]. Liu [113] and Liu et al. [84] combined a series of operations to optimize the procedure of shoreline extraction to obtain a highly accurate contourline and to avoid tedious and complex human interventions, including (a) constructing different grids on zero contour values, (b) selecting a line with a length threshold, and (c) using a smooth algorithm to smooth the shoreline in the final stage.

**The Binary Image Segmentation**

To minimize manual intervention, Liu et al. [43] proposed a substantial technical improvement through a method based on binary image segmentation (BIS) to automatically extract shorelines from airborne LIDAR data. This is the first application for extracting shorelines from airborne LiDAR by using the image-segmentation technique. It segments the DEM generated from airborne LiDAR into a binary image (land and water pixels) through the intersection of the DEM and tidal datum surface, and then extracts shorelines with spatial detail and continuity through morphologic operation. This method was quantified and evaluated based on Monte Carlo simulation, which outperforming conventional aerial photography within 4.5 m at a 95% confidence level [43].

However, Yu et al. [90] argues that it is complex and time-consuming for the filtering classification from the LiDAR point cloud and the process of the DEM generated, while using BIS to extract the shorelines. This method can only extract shoreline based on a certain tidal datum, but other auxiliary shoreline elements cannot be extracted, such as zero-meter isobaths or a low water line [90]. Zhang et al. [52] improved this method, allowing to for the extraction of a preliminary high and low tide line through the generation of a digital surface model (DSM) to intersect the surface of a water body derived from the tidal data. Considering the invalid values of DSM near the low water line (LWL) due to the LiDAR beam not being able to penetrate into water, a mobile trend surface fitting method was adopt to extrapolate DSM near LWL to extract shorelines more accurately [52]. More robust, efficient and universal techniques are proposed in their method. Meanwhile, it is also beneficial for HWL and LWL extraction, with less manual labor. However, the studies claimed there are several problems that could be investigated: (a) how to classify automatically for coastal zone and artificial coast; (b) using interpolate DSM or DEM to retain topographic features; (c) if the interest areas were large enough, it is worth discussing whether this method is suitable for extracting high and low tide lines [52].

Unfortunately, the above-mentionedmethods are unable to extract complex shorelines, or they may lead to misclassification [72]. Therefore, Yousef et al. [73] developed a multi-steps approach for using morphological operators to deal with the complex shorelines. It has the following steps: (a) using a nonparametric regression method to estimate missing elevations within the DEM data; (b) abnormality detection for reducing outliers and noises; c) utilizing a constrained morphological operations (open and close) to deal with the broken branches and land regions [72,73]. Compared to the study of Liu et al. [43], this method is easy to operate, significantly decreasing the effect from artificial structures (e.g., bridges, fishing piers, and docks) on shoreline extraction.

To sum up, although shoreline extraction methods based on DEM generated by LiDAR data promoted the development of shoreline extraction technology, there are some controversies. For example, the error of shoreline extraction would be affected by the error in generating DEM. Therefore, some shoreline extracting methods without generating DEM have been developed. For example, in order to avoid errors due to the process of generating DEM, Yu et al. [90] proposed a rasterization method to extract shorelines (Mean High Water Line), finding that those directly rasterized from LiDAR point clouds can obtain much smoother coastal terrains. The advantage of this method is the removal of high-frequency noise from point cloud data using the requantification of elevation values at the first pre-processing step. Moreover, with the demand of data precision and volume increasing, this method has simplified the process of shoreline extraction without DEM generation, especially for the point cloud data with high precision and density.

### 4.2.2. Based on an Instantaneous Shoreline

An instantaneous shoreline refers to shorelines extracted from the boundary between the water and land. One advantage of using this shoreline indicator is that it does not need to correspond to any tide-coordinated water levels [7]. Under the context, if the fusion of data from other remote sensing devices is not considered, boundary classification techniques were employed to extract the shoreline, which is similar to the segmentation of the boundary of the water and land from the pixel image. One way to extract an instantaneous shoreline is based on the intensity value classification method, and another is through the use of boundary process techniques for point clouds (Figure 2).

Intensity data is a value that represents the return strength of the laser beam. The intensity does not represent the true terrain radiance from the terrain, but it can be used for feature detection and classification [32]. With the dimensional reduction method, the 3D point cloud data are converted into a 2D image. This means that in the rasterization process, the intensity data are converted to intensity images. Compared to the processing of 3D point clouds, the processing of shoreline extraction via an intensity image is more convenient and simple. [32] developed a robust framework to mine the image of shoreline

extraction through intensity values from airborne LiDAR via cross-shore profile lines. The results were confirmed to be useful for coastal monitoring especially for the extraction and classification of coastal features. [89] proposed a fast, stable, highly accurate, and automatic extraction of shorelines based on the Alpha Shapes Model (ASM) algorithm without generating DEM. However, this method still has some defects on the regular small artificial target (e.g., the dock), which need to be further corrected using regular methods or manual editing. During the process of classification, it is also easy to misjudge the point clouds of ships on the surface to the dock, resulting in an incorrect extraction of the shoreline.

Furthermore, the ALB waveform algorithm as an alternative method for extracting shoreline in coastal zones provides support without the available tidal datum/information [80]. This paper attributes the ALB waveform algorithm to an intensity-based method, due to the waveform being a time series based on the received intensity value. Pe'eri et al. [80] compared six different waveform algorithms to distinguish land and water, and all algorithms were confirmed as having successful results. However, the best–performing algorithm was only for the IR saturation algorithm, and the worst for the red standard deviation algorithm [80]. Although these algorithms provide results with a high degree of accuracy, there are still some problems (low efficiency and inconvenience) for engineering applications [114]. Zhao et al. [114] proposed a fast and simple water-land classification using the elevation threshold interval of water surface points under the context of engineering application, which has a classification accuracy of greater than 98%. However, for the high classification accuracy requirement, the ALB waveform algorithms were still recommended to be utilized [114]. Compared to the method of shoreline extraction with the generation of DEM, the method through the use of intensity data is less well studied for shoreline extraction. However, some factors that would affect the intensity measures need to be concerned, such as variations in path length, object density, beam divergence, and others.

Although the above-mentioned methods have made a contribution to the efficiency of shoreline extraction, the extracted process may lose some information and precision in the 2D spatial environment [30]. Therefore, boundary process techniques have been developed to directly extract shoreline from point clouds data. Xu et al. [30] proposed a new series of multi-steps to extract shorelines directly from LiDAR point clouds in a three-dimensional spatial environment, which can avoid any human interaction and retain the original spatial information. As a major contribution of this method, the global minimum cost model and the use of the energy function to calculate the cost of the boundaries are proposed [30]. The results from this research had a good performance for five typical scenarios, achieving over 92.5% completeness and over 90.7% correctness [30]. Li et al. [91] puts forward an new algorithm using boundary points to segment the position of LiDAR point clouds to extract shorelines, especially for shoal and muddy environments in a coastal zone. Compared to the CLM, the result of this method using a boundary points algorithm has a better performance, with lower standard deviation (0.1656) and variance (0.0274) than CLM (0.2116 and 0.0448) [91]. The boundary points algorithm not only enriches the method of shoreline extraction, but it also has a great potential to deal with mapping different types of shorelines.

### 4.2.3. Based on Multisource Data Fusion

The complex coastal environment and human-related factors increase the difficulty of shoreline mapping. Sometimes, the real shape of the shoreline cannot be reflected by optical satellite remote sensing images, and the pixels are separated by land and ocean [115]. This problem can be avoided by using LiDAR. However, considering the limitations of airborne LiDAR, researchers no longer have to rely on airborne LiDAR as the only device to study coastal areas. LiDAR data can fuse digital images from other data sources, such as satellite remote sensing, and high-resolution or multispectral facilities (camera) equipped using airborne platform. Their aerial images or hyperspectral images (HSI) can be combined

with DEM generated from point cloud data to extract shoreline [42,72]. This allows for the combination of two-dimensional (2D) images from optical remote sensing (or data from other sensors) with three-dimensional (3D) airborne LiDAR point cloud data [55,72].

Multi-source data fusion provides support for obtaining high quality and more reliable coastal geospatial information. In the early 21st century, Lee and Shan [116], and Elaksher [42] researched coastal mapping by fusing LiDAR data and optical images (multi-spectral images or hyperspectral images). Although the focus of the research at that moment was on how to classify the elements from merging optical and laser datasets, these efforts have made contributions to updating the shoreline mapping techniques. Subsequently, applications in the coastal zone through multi-source data fusion have become a hot topic, especially for shoreline extraction [88,115,117,118].

Strictly speaking, multisource data fusion is not a specific methological concept, bacause it depends on the number of remotely sensed devices. Thus, this method is separated out with red dotted lines in Figure 2. It has an advantage in that it can flexibly select shoreline indicators according to different research purposes. For example, in order to automate the extracted shoreline from the challenge shoreline (e.g., tidal marsh and vegetated shoreline in coastal areas), Sukcharoenpong et al. [119] uses a multi-phase active contour segmentation technique to extract the shoreline, which fuses HSI and the DEM derived from airborne LiDAR data. The spectral information from the HSI can be efficiently used to obtain an accurate initialization of the active contour segmentation technique, which significantly reduces the total computational process of the method [119]. Lee et al. [48] presented a method of Mean Shift Algorithm (MSA) to extract shorelines from airborne LiDAR, which combines the color information of the aerial orthophotos. The main step of the MSA method is to classify the water surface and land in the LiDAR point clouds, based on the homogeneous characteristics of the elevation and the distributed color information of the water surface [48]. The results from the MSA method have a better performance and accuracy (0.5 m) on man-made objects than on nature shorelines (1.5 m) [48]. However, it should be noted that the shorelines extracted using the MSA are the instantaneous shorelines. In addition, the intensity data also can be combined with other remote sensing techniques to extract shorelines. Demir et al. [47] developed a method for automatically extracting lake shorelines in fuzzy clusterings, which consists of fusing Sentinel-1A SAR data with intensity values from airborne LiDAR. Its advantage is that parameters definition for the Sentinel-1A data is not required by the user, avoiding any additional interaction. To the contrary, its downside is that the quality of the data degrades when converting datasets to raster by using the mean-shift method [47].

On the other hand, fusing LiDAR and other devices provides an opportunity to use ML to extract shoreline features. Yousef et al. [72] is the first one to present a novel algorithm by using support vector machines (SVMs) to effectively classify water and land to extract shoreline by fusing DEM derived from LiDAR and their corresponding coverage of aerial images. This method allows the MHW shorelines to be extracted without reference to a tidal datum, but other shorelines (e.g., Mean Low Water) also can be extracted if the tidal datum exists [72]. Compared with the average error (4.92 m) from Yousef and Iftekharuddin [120], Lee et al. [48], and Yousef et al. [72], the SVMs method has a better performance, and the average error of the shoreline position is 2.37 m [48,72,120]. They also utilize SVM to extract shorelines from the LiDAR data that were not fused with aerial images. The results showed that the average error (2.81 m) was slightly higher than 2.37m [72]. The results are showing that SVM has a better performance and accuracy than the morphological approach based on fusing LiDAR data and its corresponding aerial images. However, the morphological method is recommended for use when data are only available from LiDAR [72]. Other literature also prove that the extraction results combining airborne LiDAR and aerial orthophoto images are more accurate than those of using individual data sources [121].

### 5. Discussion

In this paper, we reviewed publications investigating shoreline mapping using airborne LiDAR point clouds, with a particular emphasis on the research with LiDAR-derived shorelines. The most common two objectives are: extracting the features (land and water) from point cloud via segmentation and classification techniques, to acquire the instantaneous shoreline, and generating more accurate LiDAR-derived DEM to calculate the "true" shoreline using shoreline indicators. The literature demonstrates that a large number of airborne LiDAR systems have already been applied in shoreline mapping. However, there are still some limitations and challenges that need to be figured out.

Although airborne LiDAR has brought the development of shoreline research technically, it is still facing many challenges. For example, Li et al. [122] figured out that the results of shoreline extraction from airborne LiDAR are affected by some factors, such as point density, noise, reflective intensity, spatial resolution, and so on. We summarized the main factors affecting shoreline survey and extraction in three categories, including (a) objective condition limitation, (b) data availability limitation, and (c) self-characteristic limitation (Table 6).

**Table 6.** Main Limitation of Shoreline Mapping by Using Airborne LiDAR.

| Limitations | Factors | Illustration |
| --- | --- | --- |
| Objective condition | Weather condition | Fog, heavy precipitation, strong glare, and other situations [123–125] |
| | Area of interest terrain feature | Beach slope [51] |
| Data availability | Cost | Relatively expensive and depends on government-funded open data |
| | Project-based data acquisition | Low update frequency, updated data only in some areas |
| | Huge data volume | A fraction of this vast data volume is only available to be used in related studies [126] |
| Self-characteristics | Typical errors | Aircraft attitude measurements, positioning errors, IMU attitude errors, laser scanner error aperture errors, and lever arm offset errors [127] |

In the first category, several sources can affect the accuracy of airborne LiDAR surveying data. As an active remote sensing device, airborne LiDAR has a high sensitivity to weather conditions that can interfere with the laser pulses (Table 6). Especially for coastal areas, different weather condition would increase the uncertainty of absolute error for shoreline extraction. White et al. [51] stated an additional viewpoint that claims that different topographic features also increase the uncertainty of absolute error for the final shoreline level accuracy in survey sites. In addition, the accuracy and reliability of the DEM derived from airborne LiDAR may be affected by the rise of waves, where it is possible to contaminate the signals of airborne LiDAR due to the relatively gentle slopes in the intertidal zone [43].

From Section 3.2, we have figured out that data availability in coastal areas is a challenge. Although airborne LiDAR technology has been well developed over two decades and has highly reduced the cost of laser sensors due to collaboration between organizations, the cost of using airborne LiDAR data is still relatively expensive [128] (Table 6). Most data sources of the shoreline survey by using airborne LiDAR are mainly obtained from government-funded open data or regional projects (Table 4). Even with the dataset in United States, which has the largest amount of airborne LiDAR dataset, the dataset can only be updated once a year for some areas according to the official website of NOAA. In addition, the heavy equipment makes the acquisition of airborne laser scanning data difficult. For example, the total weight of the Optech CZMIL is 360 kg and the use of the

CZMIL for survey projects has specific requirements for the type of aircraft. Operational flights are also limited by the weather conditions. For all of these reasons, data availability and applicability limit the widespread use of airborne LiDAR for shoreline detection.

As for the third category, a large volume of point clouds data and survey errors are the major limiting factors (Table 6). One of the most valuable features of airborne LiDAR is the high density of the point cloud, which allows more valuable spatial information about the target to be stored [57]. Although this feature allows a huge data volume for the storing and processing of large-scale data acquisitions [129], a fraction of this vast data volume is only available to be used in related studies [126]. Moreover, the errors of airborne LiDAR data are another challenge during shoreline survey, such as typical errors [127] as shown in Table 6, which should affect the accuracy of DEM. In additon, there are still more factors that can affect the data accuracy. For example, the travel tie of laser as well as the GPS/GNSS accuracy can be affected by increasing the temperature and water vapor in the atmosphere [130].

Although these limitations exist in shoreline mapping when using airborne LiDAR, it is believed that this problem can be effectively addressed with the continuous upgrading of measuring equipment and processing methods.

The upgrade of airborne LiDAR sensors allows for improved quality of the data source. Currently, more advanced airborne LiDAR sensors are or will be released, such as single-photon LiDAR and multispectral airborne LiDAR [31], which can support and improve the shoreline extraction application. Morsy et al. [18] examined that it is possible to uses the spectral indexes from multispectral LiDAR to improve the land-water classification. Shaker et al. [33] proposed an automatic method to classify land and water via intensity data collecting using Teledyne Optech Tian. This study demonstrated that multispectral LiDAR intensity data has the ability to improve the classification results for distinguishing different land-water scenarios, including man-made shore, natural shore, shore with land depression, and an inland river [33]. Due to a lack of shoreline data, these studies only test the inland shore situations. However, they can demonstrate that multispectral airborne LiDAR can be used and enhances future coastal shoreline extraction missions. In terms of the processing methods, we find that the widely used methods to extract shorelines are based on DEM generation, especially the method of BIS and multisource data fusion. However, the point clouds data from airborne LiDAR are enormous when conducting the long-term shoreline monitoring of a large area. In this context, the current shoreline extraction methods do not look very efficient.

In recent years, ML and DL algorithms have become hot approaches in different fields and have been shown a potential capability in solving various coastal problems in the environmental and ecological fields [131], especially in dealing with the studies that require a large volume of data, such as long-term temporal and spatial scales. Some researchers have used ML to extract shoreline from optical satellite images. Bayram et al. [132] applied the Random Forest Method to extract shoreline with uncluttered results, which was a pixel-based method. However, features are required to be manually defined in ML, and the accuracy of recognition is influenced by expertise and expert experience [133,134]. Moreover, data-driven thresholds are highly customized in most methods during the process of output results, which means that these methods of shoreline extraction mentioned above cannot be generalized to all datasets under different conditions. Therefore, it is worth further discussing more efficient and convenient processing methods to extract shoreline from airborne LiDAR data, such as DL-based methods.

DL-based algorithms have been exploited as the most advanced method for studying remote sensing at present, especially for its capacity for dealing with the numerous amounts of data produced daily by airborne LiDAR. Its concept originated from the study of artificial neural networks, aiming to establish a mechanism to simulate human brain learning data features. Deep-learning methods can be utilized to any study area after the architecture has been trained [135]. In the case of shoreline extraction studies, some experts have used DL-based algorithms to extract the shoreline from optical satellite images [10,135,136].

With these foundations from optical images, it is theoretically feasible to extract shoreline from airborne LiDAR data using a DL-based algorithm. Meanwhile, considering the inherent nature of airborne LiDAR, there are still more challenges to using DL-based algorithms to extract shoreline, such as occlusions caused by blindside, noise, or unintended points, and irregular points [137]. The irregular, unstructured, and disordered nature of LiDAR point clouds are considered to be the most significant challenge in the application of DL. The irregularity of point clouds makes them uneven in the sampled scene, which leads to some areas with dense points, whereas sparse in other areas [138]. Moreover, the LiDAR point clouds have the characteristics of irregularity and disorder, which can affect the distance between two adjacent points [139].

DL-based algorithms can help researchers to effectively improve the use of large amounts of airborne LiDAR point cloud data. However, the DL-based methods usually require a large amount of training data. In the current situation, acquiring sufficient training data is still a huge challenge. However, more and more countries have started to collect or are still collecting the airborne LiDAR data on a national scale or in a coastal area, such as Japan and Canada, as mentioned in Section 3.2. Since more datasets will be available in the future, DL-based methods of shoreline extraction from airborne LiDAR could be new opportunities in coastal management.

## 6. Conclusions

The shifting and changes of shorelines has been a key issue for scientists due to the continuous occurrence of natural factors and human activity in the coastal region in recent years. Understanding the change of shoreline is very important with regard to coastal hazard assessment and integrated coastal management. Therefore, it is very important to measure and to process shoreline data accurately and efficiently. We presented a narrative review of the process of shoreline measurement and extraction, combined with airborne LiDAR during the last two decades.

First, we provide a summary of the use of laser scanning systems of airborne LiDAR to study shorelines in coastal areas, including the ALT and the ALB. In particular, the parameters of airborne LiDAR sensors used in shoreline survey are summarized and compared, which can be intuitive for knowing the development of airborne laser scanning systems used in coastal studies. We believe this summary table will be friendly in assisting those who need relevant information. This is followed by a presentation of the summary of almost all mainstream methods of shoreline extractions, including the CSP, the CLM, the BIS, multisource data fusion, and other methods combined with boundary extraction, the data of which are from the shoreline survey, using airborne LiDAR laser scanning systems. We summarize these methods into the two types of shoreline extraction methods based on our collected literature: one is the line derived from airborne LiDAR point clouds such as instantaneous shoreline, and another is the line derived from DEM generated by airborne LiDAR data, such as MHW. Additionally, we discuss the limitations of airborne LiDAR for shoreline mapping, including objective condition limitation, data availability limitation, and self-characteristic limitation, as well as the future challenges and opportunities for shoreline extraction based on DL technology.

We can note that the problem of data availability in coastal area is a main challenge that has still not been adequately solved since the limitations of the cost, region, and program projects. Another challenge is that when coastal management and research are facing the monitoring of long-term shoreline changes by using airborne LiDAR, due to its advantage of high resolution and elevation information, the shoreline will be extracted from a huge amount of point clouds data, and the processing in each temporal phase will be repeated, which does not seem very intelligent. However, these methods offer a foundation and future opportunities for extracting shoreline based on DL algorithms. In the future, the analysis and processing of the point cloud data from airborne LiDAR using DL-based algorithms should be actively explored, to improve the efficiency and quality of extracting the shoreline from LiDAR data, so that unnecessary repeated operations are reduced.

**Author Contributions:** Conceptualization, J.W. and J.L.; methodology, J.W. and L.W.; investigation, J.W., L.W, and S.F.; resources, J.W., L.W., and S.F.; writing—original draft preparation, J.W. and L.W.; writing—review and editing, J.W., L.W., S.F., S.N.F. and L.T.; visualization, J.W. and L.W.; supervision, B.P., L.H., and J.L. All authors have read and agreed to the published version of the manuscript.

**Funding:** This research received no external funding.

**Data Availability Statement:** Not applicable.

**Acknowledgments:** The first author acknowledges the China Scholarship Council (CSC) for their support via a visiting PhD scholarship. We would also like to thank the anonymous reviewers for their insightful comments and suggestions.

**Conflicts of Interest:** Conflicts of Interest: The authors declare no conflicts of interest.

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
