# Peer review of "An Overview of Shoreline Mapping by Using Airborne LiDAR"

_remotesensing, doi:10.3390/rs15010253_

Round 1

Reviewer 1 Report

The manuscript is well done and is a useful and comprehensive review of shoreline mapping by using airborne LiDAR.

The only suggestion concerns the addition of two summary tables to improve the readability of paragraphs 4 and 5, which are only textual.

Finally, in line 194 instead of the figure number there are two question marks.

Reviewer 2 Report

The paper reports an overview of methodologies commonly used for the extraction of shoreline from LIDAR data. The technical note seems to be sufficiently exhaustive with any original contribution. Nevertheless, it could represent a useful reference paper to read for a students approaching to the specific issue.

Author Response

Thank you for your comments. We have revised the manuscript, and will try our best to improve the quality of this article. 

Reviewer 3 Report

Nice review paper with sufficient number of references. It is well done.

Author Response

Thank you very much for your appreciation. We will continue to make effort on improving our research.

Reviewer 4 Report

I appreciate the opportunity to review the article titled A Review of Shoreline Mapping by Using Airborne LiDAR.  I also appreciate author’s ability to choose an insightful study, particularly one that focuses on shoreline mapping techniques which is a very dynamic part of the earth. After careful reading, I'd say the article has number of flaws which need to be fixed. Having said that, I have recommendations that I believe the authors will find useful and that, if implemented, would make this research suitable for publication in the journal.

Title

Title of the article is not explicit and somewhat unclear!! Authors need to put the basis of literature review based on which they did the research.

 Abstract

The given abstract requires a slight modification in terms of motivation and implication because it is not acceptable at its current stage. An abstract should begin with a general statement about the topic please avoid putting the given statement about definition of shoreline. Methods and possible results/outcomes are not expressed academically. The study's implication is unclear. At the end of the abstract, kindly include a statement outlining the study's implications having more explicit information regarding the main idea of this article. Do not use the term in the discussion/ we discussed in abstract.  I wonder what sort of techniques authors took for literature review, was it a systematic literature review or narrative one - not clear!!

Introduction

The Introduction section should have been improvised, a description of other studies that are related, and most crucially, the research gap that guided the development of this specific research, authors introduced this topic without proper background. I do not see the idea of the article has been established properly. Without scientific referencing and detailed analysis of those approaches, the conclusion and argument of the article do not sound appropriate. The introduction is the opening part of the scientific story to attract an audience and suggest the direction of your research. For this reason, you need to identify the problem that drives the research and introduce the key characters. If then, using the key characters, it is required to intertwine the scientific story concisely, systematically, and logically. Please rearrange the keywords so that the background of the study sounds appropriate. Having said this, I have the following observations

(1)   Shoreline change analysis requires temporal data on past shoreline How do authors explain the temporal incapability of LIDAR data for coastline extraction? Please provide some explanation in Line 53-61

(2)   I wonder if authors can put some references of Lidar mapping in regional even in national scale ( table 1)

(3)   Description on Lidar mapping across the globe is missing in the introduction sections which need to be included. A country wise LIDAR shoreline mapping is highly desirable for this study.

(4)   Please add more citation if required as I see many sentences do not have citations, for example 62-64.

 Other recommendations are given below

 (1)   I cannot see much information on Evolution of Airborne LiDAR for shoreline mapping in this section rather history of Lidar techniques are provided 88-109

(2)   Information can be shorten for the two sections namely Airborne laser topographic and  bathymetric scanning system for shoreline measurement

(3)   Figure 2 and 3 are not explicit consider replacing with other relevant figures

(4)   Title of table 2 is not clear

(5)   It is expected that a technique section will give users a clear understanding of how relevant articles were extracted and sorted. Without a proper method section reader will not comprehend the outcomes of the research

(6)   There is not result and discussion section for this article which is integral part of a scientific article. Please reorder the sections and put them in the Result sections cause at its present form it is hard to understand the finding of the study. As per as the discussion section is concern   This article does not have a proper scientific discussion section from where the reader can get comparative discussion on data and techniques relative to coastal dynamics. If possible please elaborate limitation, future direction and policy implication of this study

(7)   The write-up and design of the demonstration are often ambiguous because the entire article did not adhere to the format of a scientific article. Citation information and figure numbers are absent in the text. Figure 4 is not a good presentation of the topic. There are ways to make this flow chart better. Writing needs to be improved and should be more succinct in order to provide a clear and distinct idea to readers.

(8)    Conclusion should be revised, too much of generic sort of discussion here, I cannot see the brief of results of this study here. Please revise conclusion according. 

Reviewer 5 Report

Comments for Remote Sensing,

The aim of the manuscript is to provide a comprehensive review of shoreline mapping. It is regular review. It can be published on Remote Sensing although it could be more concise and be better organized.

First, the authors did not distinguish two types of shorelines: 1) a line derived from lidar itself such as instantaneous water line, and 2) a line derived from DEM by another factor such as MHW.  The MHW/MLW and other this kind lines are not relevant to lidar. For a region, MHW/MLW are obtained by ocean observation network. We can use a MHW value to get a contouring line from a DEM derived from any data source such as radar. How we get the optimal DEM from lidar point clouds is a general topic of airborne lidar, it is not restricted to for shoreline mapping.

Second, shoreline such as instantaneous water line and vegetation line are results of lidar point classification. In other words, a line separating land and water points can be explained as instantaneous water line for a coastal zone. And vegetation line is a line marking the seaward limit of vegetation growth.  A vegetation line can be identified when we know open water, soil, and vegetation. However, lidar point classification is the main topic of airborne lidar.

Third, Section 5 are not closely related to shoreline mapping. Of course, advances in lidar point clouds segmentation will benefit to extract shoreline. It is common sense. If the authors focused on special techniques of shoreline mapping, the review could be more concise and more helpful to the coast community.

Round 2

Reviewer 4 Report

Happy with the corrections